# Microscopic origins of conductivity in molten salts unraveled by computer simulations

Marie-Madeleine Walz [1] & David van der Spoel [1]✉

Molten salts are crucial materials in energy applications, such as batteries, thermal energy storage systems or concentrated solar power plants. Still, the determination and interpretation of basic physico-chemical properties like ionic conductivity, mobilities and transference numbers cause debate. Here, we explore a method for determination of ionic electrical mobilities based on non-equilibrium computer simulations. Partial conductivities are then determined as a function of system composition and temperature from simulations of molten $LiF_\alpha Cl_\beta I_\gamma$ (with $\alpha + \beta + \gamma = 1$). High conductivity does not necessarily coincide with high $Li^+$ mobility for molten $LiF_\alpha Cl_\beta I_\gamma$ systems at a given temperature. In salt mixtures, the lighter anions on average drift along with $Li^+$ towards the negative electrode when applying an electric field and only the heavier anions move towards the positive electrode. In conclusion, the microscopic origin of conductivity in molten salts is unraveled here based on accurate ionic electrical mobilities and an analysis of the local structure and kinetics of the materials.

[1] Department of Cell and Molecular Biology, Uppsala University, Husargatan 3, Box 596, SE-75124 Uppsala, Sweden. ✉email: david.vanderspoel@icm.uu.se

Many different materials have been proposed as electrolytes for, e.g., battery applications, from the simplest molten salts[1,2] to combinations of anionic organic polymers with metal cations[3,4]. A key design objective for electrolyte materials is high conductivity[1,2,5]. Therefore, it is cumbersome that the interpretation and origin of trends seen in conductivity data remain a topic of unsettled debate[6–11]. One issue is, whether the existence of ion association, such as ion pairing and clustering, is needed to explain conductivity values that are lower than expected, if one assumes that the conductivity is completely determined by diffusive motion of uncorrelated ions. However, such trends can be interpreted without the need for ion association[6,8,9,12,13] and be formally explained by using velocity cross-correlation coefficients[6]. Still, ion association does exist in many ionic liquids (ILs) and molten salts[8,14,15], and may cause a deviation from the ideal conductivity. In this work, we quantify the partial conductivities in relation to ion association using accurate simulation models[16] and an analysis of ion dissociation kinetics[17].

A high conductivity implies many charge carriers per time (electric potential and length), which can be realized through either (1) a high ionic electrical mobility and/or (2) a high number density, more so if in combination with low ion cross-correlation effects. An increase of the temperature is an obvious manner to increase the velocity of the ions, which in turn increases the conductivity in general. However, with increasing temperature, the overall liquid expands, whereas the counter-ions get closer together[17,18] and the energy barrier for hopping to the next neighbor increases[9,12,17], which can lead, in certain cases, even to a maximum in the conductivity as a function of temperature such as, e.g., in the case of semicovalent molten halides (e.g., $BiCl_3$)[9,12,13,19].

Molten salts that contain $Li^+$ ions are of high relevance in the energy industry, e.g., in liquid metal batteries, where such salts provide high ionic conductivity, resulting in a high rate capability and energy efficiency[1,2,5,20]. It is commonly assumed that a high ion conductivity coincides with a high ion mobility[2,5,21]. For instance, Masset and Guidotti[21] reason that lithium-based electrolytes exhibit the highest ionic conductivities due to the fact that the mobility of the lithium cation is higher than that for other alkali-based electrolytes. In another example, Li et al.[2] write that, in general, the mobility of ions in molten salts follows Arrhenius-type behavior and, therefore, that a higher mobility of the ions implies better ionic conductivity of a molten salt. Such statements do not hold in general, as we show in this contribution.

Using state-of-the-art equilibrium and non-equilibrium computer simulations of molten salts of the general formula $LiF_\alpha Cl_\beta I_\gamma$ with $\alpha + \beta + \gamma = 1$ (Table 1), we show that the highest conductivity not necessarily coincides with the highest $Li^+$ electrical mobility. Furthermore, the ionic electrical mobilities are investigated by means of non-equilibrium simulations where an external electric field was applied[22–24]. Partial conductivities were determined from the partial mobility values; it is challenging to determine these properties experimentally[14,15] and they often are approximated based on diffusion coefficients, which, as shown below, is quantitatively and qualitatively incorrect.

For the molecular dynamics (MD) simulations, we used a force field for alkali halides[16], which was recently introduced as part of the Alexandria project[16,25,26]. The model uses buffered van der Waals interactions[27] and Gaussian-distributed charges combined with explicit polarization. Simulations including explicit polarizability of the ions effectively model three body interactions, going beyond the pair potential approximation underlying classical force fields[28]. It has been shown that the model predicts physicochemical properties accurately over a wide range of temperatures for the gas, liquid, and solid phases, compared to both other force fields and experimental data[16–18,29].

## Results

**What causes a high conductivity?** It is well-established that accurate conductivity values can be determined from the current autocorrelation function (ACF) obtained in MD simulations, when a Green–Kubo (GK) relation is used. The current ACF directly includes cross-correlation effects between ions[7,30]. We evaluate the GK conductivity using an implementation developed by Dommert et al.[31] for the analysis of ILs. For comparison, we also determined the conductivities using the Einstein–Helfand (EH) method[32] and the results agree well with the GK method (Supplementary Note 1 and Supplementary Tables 1 and 2). The electrical mobility values $b_{EF}$ were evaluated using non-equilibrium simulations; for this purpose, external electric fields $E$ of varying field strengths (in the linear response regime) were applied and the mobilities $b_{EF}$ were taken to be the slope of a plot of the ion drift velocities $v_d$ as a function of the electric field $E$ (see Supplementary Methods and Supplementary Fig. 1). For simplicity, we will refer in the discussion below to a positive and negative electrode rather than an electric field; however, it is crucial to remember that there are no "physical" electrodes obstructing the flow of the ions in the simulation. Figure 1 shows the GK conductivity vs. the $Li^+$ mobility and vs. the $Li^+$ number density. Two opposite trends can be observed, both of which affect the conductivity. We consider two separate cases as follows:

1. studies of the same system ($LiF-LiCl-LiI_{eut/400/450}$) at different temperatures (Fig. 1a, c). Here, an increase in conductivity results from an increase in $Li^+$ mobility that is accompanied by a drop in the $Li^+$ number density.
2. different $LiF_\alpha Cl_\beta I_\gamma$ compositions at the same temperature (Fig. 1b, d). In this case, an increase in conductivity is based on an increase in $Li^+$ number density, even though this in fact is accompanied by a drop in the $Li^+$ mobility. That means, in this case, a higher mobility does not correlate with higher conductivity.

For the range of systems and temperatures investigated, this implies that a lower number density correlates with faster $Li^+$ mobility independent of the cause, be it a change in temperature or the systems composition.

The data points for the $LiF_\alpha Cl_\beta I_\gamma$ systems are plotted in ternary diagrams to visualize the trends described for case (2). Figure 2 shows the $Li^+$ electrical mobility $b_{EF,Li^+}$ (a), the $Li^+$ number density $\rho_{N,Li^+}$ (b), the GK conductivity $\sigma_{GK}$ (c), and the Haven ratio $H$ ($=\sigma_{NE}/\sigma_{GK}$) (d) for the $LiF_\alpha Cl_\beta I_\gamma$ mixtures at different compositions, all at 1200 K. Here, $\sigma_{NE}$ is the conductivity computed using the Nernst–Einstein (NE) relation (Supplementary Note 1 and Supplementary Table 1), which uses the diffusion coefficients, thus neglecting ion cross-correlation effects, to

**Table 1 Compositions of binary and ternary mixtures.**

| Electrolyte | Composition/mol% |
|---|---|
| LiF-LiCl$_{eut}$ | 30.5-69.5 |
| LiF-LiI$_{eut}$ | 16.5-83.5 |
| LiCl-LiI$_{eut}$ | 34.6-65.4 |
| LiF-LiCl-LiI$_{eut}$ | 11.7-29.1-59.2 |
| LiF-LiCl-LiI$_{400}$ | 20-40-40 |
| LiF-LiCl-LiI$_{450}$ | 25-55-20 |

*eut* eutectic.
The following binary and ternary salt mixtures containing LiF, LiCl, and/or LiI are studied, besides the pure LiF, LiCl, and LiI. The subscript 400 and 450 in the ternary mixtures indicates the approximate melting point in degree celsius.

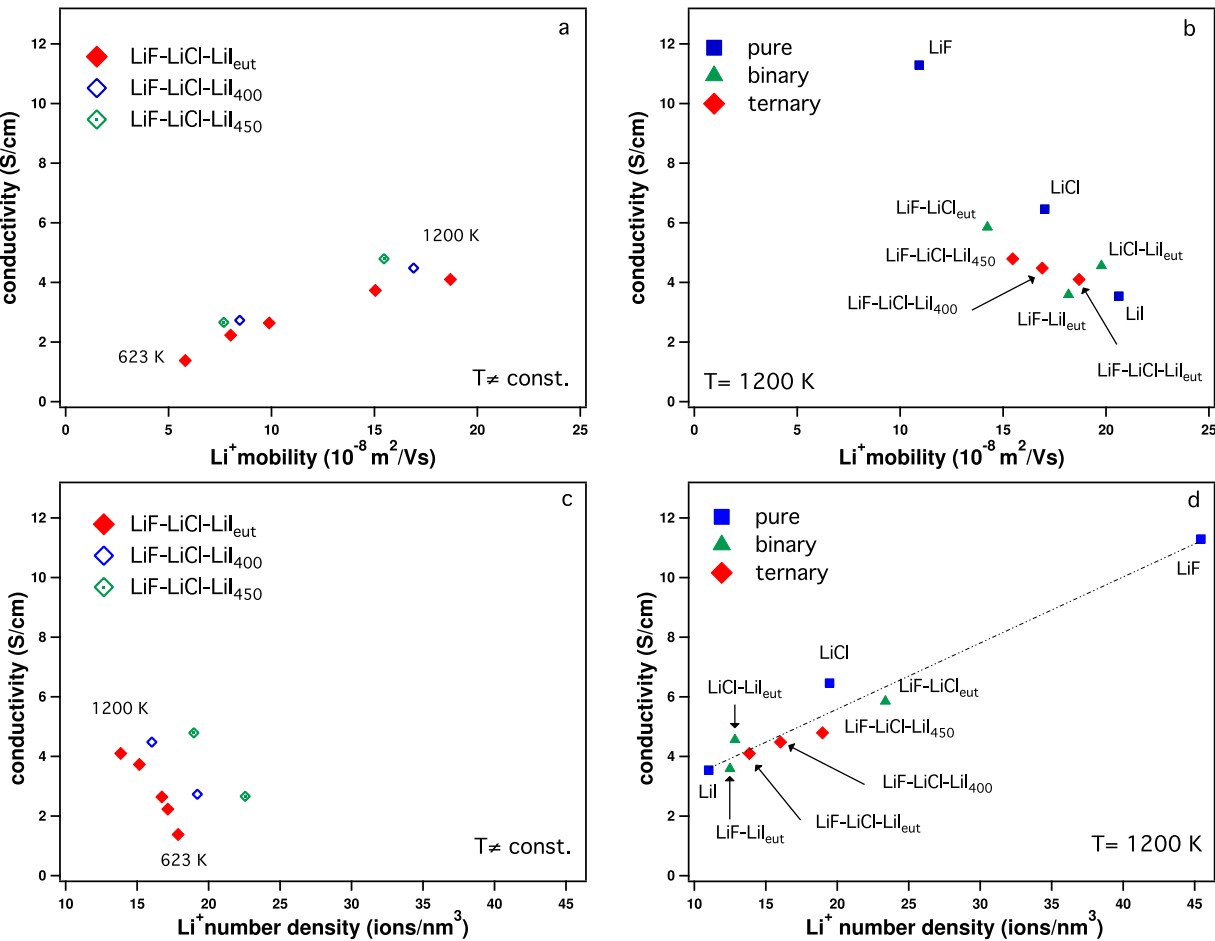

**Fig. 1 Conductivity as a function of Li$^+$ mobility and number density.** Conductivity vs. Li$^+$ mobility (**a**, **b**) and vs. Li$^+$ number density (**c**, **d**). **a**, **c** At different temperatures for LiF-LiCl-LiI$_{eut}$ (623, 723, 773, 1000, and 1200 K) and for LiF-LiCl-LiI$_{400/450}$ (773 and 1200 K), a higher mobility and lower number density correspond to a higher temperature. **b**, **d** At the same temperature for various all-Li$^+$ systems (LiF, LiCl, LiI, LiF-LiCl$_{eut}$, LiF-LiI$_{eut}$, LiCl-LiI$_{eut}$, LiF-LiCl-LiI$_{eut}$, LiF-LiCl-LiI$_{400}$, and LiF-LiCl-LiI$_{450}$) at 1200 K, the mobility and number density are a function of the system composition. Numerical details are provided in Supplementary Note 1 and Supplementary Tables 1 and 2.

estimate an upper bound of the conductivity. The diffusion coefficients were computed using velocity ACFs in equilibrated systems and, for comparison, also using the mean square displacement method (Supplementary Note 1 and Supplementary Table 1). A high Haven ratio signifies that ion cross-correlation effects reduce the conductivity far below its theoretical upper bound. As $b_{EF,Li^+}$, $\rho_{N,Li^+}$, and $\sigma_{GK}$ vary slowly, nine compositions are sufficient to show the trends. Interestingly, the highest conductivity is obtained in pure LiF, whereas the highest $b_{EF,Li^+}$ is found in pure LiI. The reason for this apparent contradiction is that conductivity depends on both the ion's mobility and on the material's number density (Fig. 2a, b), i.e., for the different LiF$_\alpha$Cl$_\beta$I$_\gamma$ systems at the same temperature, a high conductivity originates from a high number density. The amount of ion cross-correlation, reducing the ideal conductivity, depends on the species under investigation, in this case the anion. High values for the Haven ratio are observed for LiI, LiF-LiI$_{eut}$, LiF-LiCl-LiI$_{eut/400/450}$, and LiCl-LiI$_{eut}$, all of which contain large amounts of the heavier anions, Cl$^-$ and/or I$^-$.

Masset et al.[5] observed that the LiF-LiCl-LiI$_{eut}$ electrolyte behaves differently from other Li$^+$-containing electrolytes: they note that despite that it only contains lithium as cations (all-Li$^+$), it exhibits a conductivity that is lower than other all-Li$^+$-electrolytes. Masset et al.[5] suggest that the lower conductivity of the electrolyte LiF-LiCl-LiI$_{eut}$ could be explained by its high density compared to other electrolytes. From our ternary contour plot, it is clearly visible that LiF-LiCl-LiI$_{eut}$ indeed has a low conductivity, but that it is part of a continuous change dependent on the composition. Rather, we conclude that the low conductivity in this region is due foremost to the low Li$^+$ number density (which, as mentioned previously, is accompanied by a high Li$^+$ mobility) and increased ion cross-correlation effects.

In analogy to the empirical equation that was derived by Redkin et al.[33] to describe the conductivity in dependence of temperature and composition, we can interpolate our conductivity data with the following parametric expression:

$$ln(\sigma_{x_i,T}) = \alpha - \frac{\beta}{T} + \frac{\gamma}{x_{LiF}\delta + x_{LiCl}\epsilon + x_{LiI}\zeta} + \frac{\eta}{(x_{LiF}\delta + x_{LiCl}\epsilon + x_{LiI}\zeta)^2}$$

(1)

with $T$ being the melt's temperature, $x_i$ being the molar fraction of the melt's components, and $\alpha$, $\beta$, $\gamma$, $\delta$, $\epsilon$, $\zeta$, and $\eta$ being the parameters that have been fitted to reproduce the conductivity data. Using this parametric expression, all simulated conductivity data (for different compositions and temperatures) can be reproduced with an root-mean-square deviation value of 0.2 S cm$^{-1}$. For the calculated conductivity values and the fitted parameters, the reader is referred to Supplementary Note 4 and Supplementary Table 4.

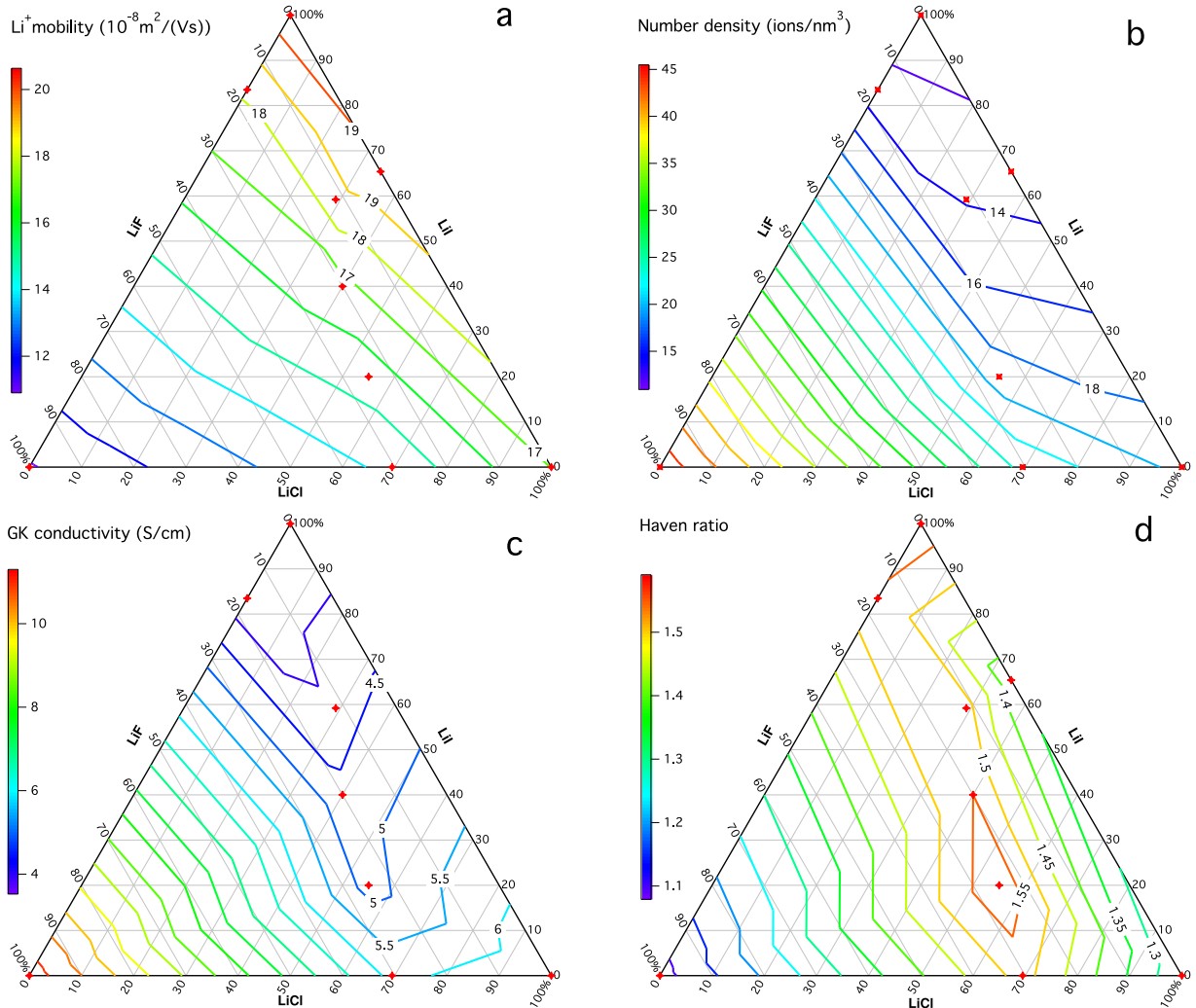

**Fig. 2 Composition-dependent properties for LiF$_\alpha$Cl$_\beta$I$_\gamma$ at 1200 K.** Ternary diagram of **a** Li$^+$ electrical mobility in $10^{-8}$ m$^2$ V$^{-1}$ s$^{-1}$, **b** Li$^+$ number density in ions per nm$^3$, **c** GK conductivity in S cm$^{-1}$, and **d** Haven ratio $= \sigma_{NE}/\sigma_{GK}$ for salts containing LiF, LiCl, and/or LiI at 1200 K. $\sigma_{NE}$ was calculated using the diffusion coefficients that were determined by means of the velocity autocorrelation function (vac). A higher number of the Haven ratio indicates more ion cross-correlations (it does not necessarily indicate more ion association)[6]. Calculated points are shown as red diamonds; the contour lines are linearly interpolated.

**How do the ions contribute to the total conductivity?** Table 2 compares partial ionic electrical mobility and partial conductivity values for LiF-LiCl-LiI$_{eut}$ (see Table 1 for composition) at different temperatures (all other values for the investigated pure, binary, and ternary molten salts are listed in Supplementary Note 4 and Supplementary Table 3). Two types of mobility values are compared: (1) $b_{EF}$, determined by means of applying an external electric field. (2) $b_D$, determined from diffusion coefficients, using $b_i = (D_i q_i)/(k_B T)$[34]. As explained above, the $b_D$ values neglect ion cross-correlation effects and $b_D$ equals $b_{EF}$ only in case there are no interactions between the ions, e.g., in very dilute solutions[14]. Ion cross-correlation effects can be quantified by evaluating diffusion coefficients $D_{ij}$ from the slope of the long time correlated displacements of the species $i$ and $j$. Zhang et al.[24] uses $D_{ij}$ to calculate $\sigma_{--}$, $\sigma_{++}$, and $\sigma_{+-}$, which represent the conductivity contributions arising from the anion–anion, cation–cation, and cation–anion correlated motions, with the total conductivity defined as $\sigma = \sigma_- + \sigma_+ + \sigma_{--} + \sigma_{++} + 2\sigma_{+-}$ where $\sigma_-$ ($= (e^2/k_B T)\rho_- z_-^2 D_-$) and $\sigma_+$ ($= (e^2/k_B T)\rho_+ z_+^2 D_+$) are the conductivity contributions arising from the anions and cations that move in an uncorrelated manner.

The sign of the mobility values $b_{EF}$ corresponds directly to the direction in which the ions drift when applying an external electric field. The $v_d$ and hence $b_{EF}$ are evaluated within the center-of-mass frame of reference and with periodic boundary conditions. Due to conservation of momentum, the drift velocity (and hence mobility) of one of the components is not an independent variable.

According to textbook knowledge, positively charged cations travel towards the negatively charged electrode, while the anions drift in the opposite direction, i.e., towards the positively charged electrode. $b_D$ values are by definition consistent with this picture. This behavior is observed as well for $b_{EF}$ in the pure molten salts (Supplementary Note 4 and Supplementary Table 3): a positive value for the cation and negative value for the anion. In contrast, the $b_{EF}$ values for mixtures show a different trend. In LiF-LiCl-LiI$_{eut}$, the lighter anions, F$^-$ and Cl$^-$, move on average in the same direction as Li$^+$, i.e., towards the negatively charged electrode, and solely the heavier I$^-$ drifts towards the positively charged electrode (Supplementary Movies 1 and 2). The same trends are observed for the binary and other ternary mixtures (Supplementary Note 4 and Supplementary Table 3), where the

**Table 2 Temperature-dependent electrical mobility and conductivity data for LiF-LiCl-LiI$_{eut}$.**

| T/K | M | $b_{Li^+}$ | $b_{F^-}$ | $b_{Cl^-}$ | $b_{I^-}$ | $\sigma_{Li^+}$ | $\sigma_{F^-}$ | $\sigma_{Cl^-}$ | $\sigma_{I^-}$ | $\sigma_{tot}$ | H | $\sigma_{GK}$ | $\sigma_{exp}$ |
|---|---|---|---|---|---|---|---|---|---|---|---|---|---|
| 623 | EF | **5.8** | **2.8** | **0.2** | **−0.6** | **1.7** | **−0.1** | **−0.02** | **0.1** | **1.7** | 0.9 | 1.4 | 2.2 |
|  | D | 3.5 | −1.5 | −1.7 | −1.7 | 1.0 | 0.05 | 0.1 | 0.3 | 1.4 |  |  |  |
| 723 | EF | **8.0** | **4.4** | **0.5** | **−0.9** | **2.2** | **−0.1** | **−0.04** | **0.2** | **2.2** | 1.1 | 2.2 | 2.7 |
|  | D | 5.9 | −2.9 | −3.2 | −3.2 | 1.6 | 0.1 | 0.3 | 0.5 | 2.5 |  |  |  |
| 773 | EF | **9.9** | **5.3** | **0.4** | **−1.1** | **2.7** | **−0.2** | **−0.03** | **0.2** | **2.6** | 1.1 | 2.6 | 2.9 |
|  | D | 7.1 | −3.7 | −4.1 | −4.1 | 1.9 | 0.1 | 0.3 | 0.7 | 3.0 |  |  |  |
| 1000 | EF | **15.0** | **7.8** | **1.1** | **−1.8** | **3.7** | **−0.2** | **−0.08** | **0.3** | **3.6** | 1.4 | 3.7 | 3.7 |
|  | D | 12.8 | −8.0 | −7.9 | −7.8 | 3.1 | 0.2 | 0.6 | 1.1 | 5.0 |  |  |  |
| 1200 | EF | **18.7** | **10.6** | **2.0** | **−2.3** | **4.1** | **−0.3** | **−0.1** | **0.3** | **4.0** | 1.5 | 4.1 | 4.3 |
|  | D | 17.3 | −10.3 | −12.0 | −10.3 | 3.8 | 0.3 | 0.8 | 1.4 | 6.2 |  |  |  |

Ion mobilities ($10^{-8}$ m$^2$ V$^{-1}$ s$^{-1}$), (partial) ion conductivities (S cm$^{-1}$), and Haven ratios of LiF-LiCl-LiI$_{eut}$ at different temperatures. Method (M) for evaluation of mobility $b$ and conductivity $\sigma$ values: (1) (EF, in bold) via non-equilibrium MD simulations applying an electric field or (2) (D) via equilibrium MD simulations using diffusion coefficients determined from the velocity autocorrelation function; for the latter, the mobilities are approximated using $b_i = (D_i q_i)/(k_B T)$. The partial conductivity values were calculated using $\sigma_i = \rho_{N,i} q_i b_i$ with $\rho_{N,i}$ being the number density, $q_i$ being the charge, and $b_i$ the mobility. The total conductivity is $\sigma_{tot} = \sum_i \sigma_i$; the conductivities $\sigma_{EF}$ and $\sigma_{NE}$ are listed under $\sigma_{tot}$, where $\sigma_{EF}$ is calculated using the mobility values from the EF simulations and $\sigma_{NE}$ using the mobility values determined from $D_{vac}$ (Supplementary Table S2). The Haven ratio is here defined as $H = \sigma_{NE}/\sigma_{EF}$. For comparison, the GK and the experimental conductivity are listed. The error of the conductivity is estimated to be ~5%.

one or two lighter anions travel on average in the same direction as the Li$^+$, whereas the other heavier one(s) drift(s) in the opposite direction, depending on the composition of the mixture. This observation that seems, at a first glance, to defy the simplistic picture of independent ions in an electric field is in fact in line with electrostatics as the ions are strongly interacting with each other. The average drift direction of the lighter anions is in agreement with Drude theory (see below) and in line with the formation of overall positively charged short-lived clusters (see below).

The partial conductivities were calculated from the partial ionic electrical mobility values via $\sigma_i = \rho_{N,i} q_i b_i$, with $\rho_{N,i}$ the number density, $q_i$ the charge, and $b_i$ the mobility. The total conductivity is given by $\sigma_{tot} = \sum_i \sigma_i$[35]. The non-equilibrium conductivity values $\sigma_{EF}$ are in general very similar to the independently derived $\sigma_{GK}$ reference values (see Supplementary Note 4 and Supplementary Table 3 for other investigated systems) and to experimental data. This demonstrates that the effective electrical mobility values $b_{EF}$ take cross-correlation effects into account correctly. For the pure molten salts (see Supplementary Note 4 and Supplementary Table 3), Sundheim's universal golden rule, $\sigma^+/\sigma^- = m^-/m^+$[36,37], obtained from the generalized Drude theory as a law of motion under an electric field[22,23,37], is found to hold true. The rule can be extended to binary and ternary mixtures[38]; in our case, $\sigma^+ m^+ = \sum_i m_i^- \sigma_i^-$. The partial conductivity values from our binary and ternary mixtures, determined from non-equilibrium simulations, are in line with this golden rule as well.

It is possible to define partial Haven ratios from the partial conductivities with $H_i = \sigma_{i,NE}/\sigma_i$ using the conductivity that is calculated from the $b_D$ values, which equals the NE conductivity, $\sigma_{NE}$ (Table 2). Some authors have used this to investigate which ion's conductivity deviates most from the NE partial conductivity due to ion cross-correlation effects[15,39–41]. However, in mixtures, the partial conductivities, for instance observed for $\sigma_{Cl^-}$ (see Table 2), can be close to zero, which would yield the partial Haven ratio undefined. Table 2 also highlights that the extent of ion cross-correlation, as defined from the total Haven ratio, is temperature-dependent.

The simulations underestimate the experimental conductivity, $\sigma_{exp}$, somewhat at low temperature (see Table 2, 623 K), which can be explained by considering that the melting point is a difficult material property to predict ($T_{m,exp}$(LiF-LiCl-LiI$_{eut}$) = 614 K[21]). In our recent investigation of melting points of pure alkali halides[29], we found a root-mean-square deviation of about 80 K, with Li-salts being the most difficult ones to predict. Although our predictions using the Alexandria alkali halide force field[16] are a factor of three more accurate than earlier work, it suggests that calculations near the melting point may be less accurate than those at higher temperature. At temperatures relevant for applications (around 773 K), the agreement is satisfactory.

Transference numbers for all investigated systems are listed in Supplementary Note 4 and Supplementary Tables 5 and 6. If we choose to define the transference number as $t_i = \sigma_i/\sum_j \sigma_j$[14,35,39,42], it is a measure for the fractional contribution of each ion to the total conductivity. It should be noted that the sign of the partial conductivities $\sigma_{i,EF}$ (and thus the sign of $t_i$) is the result from the direction of the average drift velocities of the ions in the applied electric field, that are direct observables. Due to the conservation of momentum there are $N − 1$ independent observable velocities for $N$ components, i.e., one of the variables is dependent on the others, in a periodic system. If there are additional boundary conditions, such as impenetrable electrodes, the number of independent variables may be reduced further[35]. The authors note that there is an ongoing debate regarding transference numbers of pure salts and mixtures; in particular, which frame of reference should be chosen and whether resulting transport numbers can be interpreted in terms of ion association[10,11,43,44]. In this work, the $t_i$ are used strictly as a measure of the fractional contributions to the total conductivity; the interaction of the ions will be discussed separately below. We find that Li$^+$ accounts for between 83% and almost 100% of the conductivity in all systems, except pure LiF (73%). Transference numbers are often approximated using partial NE conductivities[14,39], which for the here investigated systems would result in that Li$^+$ contributes 53–67% to the conductivity, which significantly underestimates the true values. The anions have minor contributions only, which may even cancel each other, as their contributions to the total conductivity may be either negative or positive. Due to its low mobility, Cl$^-$ has the lowest contribution to the conductivity, despite the fact that it is not the smallest molar fraction. From a comparison of transference numbers determined using the EF and the NE partial conductivities (Supplementary Tables 5 and 6), it is clear that $t_i^{NE}$ values are both quantitatively and qualitatively misleading.

Gouverneur et al.[14] studied Li-salts dissolved in different ILs (LiTFSA/EmimTFSA and LiBF$_4$/EmimBF$_4$) using electrophoretic nuclear magnetic resonance to investigate mobilities and partial conductivities. These authors observed that Li$^+$ and the anion have the same drift velocity, and infer from their data that Li$^+$ formed clusters with anions, leading to a diffusion of those overall negatively charged Li-containing clusters towards the positively

**Table 3 Ion-pair lifetimes, thermodynamic and structural data of $LiF_\alpha Cl_\beta I_\gamma$ at 1200 K.**

| Ion pair | $\tau$ | | | $\Delta^\ddagger G$ | | | $N$ | | | $r$ | | |
|---|---|---|---|---|---|---|---|---|---|---|---|---|
| Salt | LiF | LiCl | LiI | LiF | LiCl | LiI | LiF | LiCl | LiI | LiF | LiCl | LiI |
| LiF | 1.5 | | | 36.3 | | | 5.1 | | | 186 | | |
| LiCl | | 1.7 | | | 37.6 | | | 4.8 | | | 236 | |
| LiI | | | 2.2 | | | 40.2 | | | 4.4 | | | 272 |
| LiF-LiCl-LiI$_{eut}$ | 4.3 | 2.4 | 2.0 | 46.6 | 40.9 | 39.1 | 3.5 | 4.3 | 4.8 | 180 | 232 | 276 |

Lifetimes $\tau$ (ps), Gibbs free energies of activation $\Delta^\ddagger G$ (kJ mol$^{-1}$) of bond breaking, coordination numbers $N$, and interionic distances $r$ (pm) for the ion pairs in LiF-LiCl-LiI$_{eut}$ and the pure salt melts at 1200 K (evaluated from equilibrium simulations). The coordination numbers for LiF-LiCl-LiI$_{eut}$ in the table reflect the number of Li$^+$-ions around one anion. In LiF-LiCl-LiI$_{eut}$, one Li$^+$-ion has at 1200 K on average 0.4 F$^-$, 1.2 Cl$^-$, and 2.9 I$^-$ around itself, i.e., in total 4.5 anions; implying a mixture of fluctuating coordination geometries over space and time, e.g., four- and fivefold coordination geometries (see Supplementary Movie 1).

charged electrode. The experimental observations by Gouverneur et al.[14] have been replicated using MD simulations[42]. In the molten Li-halide salts studied here, the cation and anions do not have the same mobility values, but instead drift at very different speeds, which is in line with the observed short ion-pair lifetimes on the ps-timescale (see below)[17]. Nevertheless, we find that the lighter anions drift in the "wrong" direction, such as the cation Li$^+$ is observed to do in the study by Gouverneur et al.[14]. The observation in our study that anions drift in the "wrong" direction, whereas in Gouverneur's study the cation does, is based on the fact that here mixed systems with the same cation are studied, whereas Gouverneur et al.[14] investigate mixed systems with the same anion. The systems studied by Gouverneur et al.[14] were all at room temperature, which may imply higher activation energy of Li$^+$-hopping[17].

In another study, Gheribi et al.[15] investigated partial conductivities in molten NaF-AlF$_3$ with different compositions from equilibrium MD simulations using a partition ansatz. For this purpose, the authors postulate the relative contribution of each ion to the cross-correlation. Based on the partial conductivities, they claim that the Al$^{3+}$ ion travels on average in overall negatively charged clusters, AlF$_n^{3-n}$ with $n > 3$, towards the positively charged electrode. However, we noticed that the partial conductivity values determined by Gheribi et al.[15] are not compatible with the universal golden rule[36,37], even for the simplest case, i.e., pure NaF, which is also an alkali halide such as the pure LiF, LiCl, and LiI, investigated here without any need for assumptions. Indeed, Koishi et al.[23] showed that the universal golden rule holds for molten NaCl as well. This strongly suggests that direct determination of the partial conductivities from the calculated electrical mobility values $b_{EF}$ is to be preferred over the approach by Gheribi et al.[15,39,45]. It should be noted that negative effective transference numbers for cations have been reported in a few additional publications[46–49]. This is, to the best of our knowledge, the first time negative transference numbers for anions are reported. It may be clear that effective partial conductivities are system, composition, and temperature-dependent.

**Microscopic view of the conductivity.** All ions have different electrical mobilities, i.e., they travel at various speeds in the electric field (Table 2). Ion pairs and clusters are formed, and the Li$^+$ seem to effectively drag along the lighter anions. Clearly, when the anions drift in the electric field direction similar to Li$^+$, the formed clusters must be overall positively charged, to allow the movement of the negatively charged ion in the direction of the electric field. In general, molten alkali halides are considered to be fully ionized[13,50,51], i.e., no local coordination polyhedra are thought to form in these ionic systems. However, even for such simple molten salts, intermediate range chemical ordering for different alkali cations with a common anion has

been observed in different Li-halides mixed with other alkali halide salts[51,52]. Although both experimental and computational studies on the local structures of alkali halide mixtures with a common anion exist[51–54], there are none concerning mixtures with a common alkaline[55], such as the mixtures investigated here.

Radial distribution functions (Supplementary Note 5 and Supplementary Fig. 9) show that the cation–anion distances in LiF-LiCl-LiI$_{eut}$ decrease with increasing temperature, i.e., the counter-ions get closer together[17,18]. When comparing LiF-LiCl-LiI$_{eut}$ with the pure compounds at the same temperature, it can be seen that the Li$^+$-F$^-$ and Li$^+$-Cl$^-$ distances are shortened in the eutectic mixture (−6 and −4 pm), whereas the Li$^+$-I$^-$ distance is slightly longer (+4 pm) (see Table 3). In comparison to the pure salt melts, these changes in the bond distances lead to lower coordination numbers (CNs) of Li$^+$ around the anions F$^-$ and Cl$^-$, and a higher CN of Li$^+$ around I$^-$ in the eutectic mixture. These structural trends are consistent with the observation that the lighter anions interact more strongly with the cation and therefore drift in the same direction as the Li$^+$ ion.

The directionality of the Li$^+$-anion bond vectors in the simulation box was analyzed and, as expected, no preferred orientation of the bond vectors is observed in the absence of an electric field. However, in the presence of a field, the anions F$^-$ and Cl$^-$ bound to a Li$^+$-ion are, on average, preferentially located towards the positive electrode, whereas I$^-$ is slightly preferentially located towards the negative electrode. This stereoisomerism can be rationalized in terms of the average drift direction of the anions in the electric field. As I$^-$ is drifting towards the positive electrode and Li$^+$ towards the negative electrode, the cation bumps into I$^-$, whereas on the opposite side of the Li$^+$-ion and I$^-$ may leave the coordination shell. In contrast, F$^-$ and Cl$^-$ are drifting in the same direction as Li$^+$, but with a lower velocity, leaving them on the opposite side of the coordinated I$^-$ anion(s) (Supplementary Movie 2). The magnitude of the directionality is dependent on the electric field strength, i.e., the larger the electric field strength, the more pronounced is the preferential orientation of the anions around the cation; the radial distribution functions, however, remain virtually unchanged.

The ion-pair lifetimes $\tau_{LiF}$ and $\tau_{LiCl}$ are longer in the eutectic mixture than in the pure compounds (Table 3). In contrast, $\tau_{LiI}$ is shorter in the eutectic mixture than in pure LiI. These lifetimes can be converted to Gibbs free energies of activation $\Delta^\ddagger G$ for the breaking of the ionic bonds if one assumes bond breaking can be approximated as an Eyring process (see Supplementary Note 6 and Supplementary Eq. (8))[56]. Li$^+$-F$^-$ and Li$^+$-Cl$^-$ have higher activation barriers for bond breaking in the eutectic mixture than in pure compounds, whereas for the Li$^+$-I$^-$ bond it is slightly lower. Interestingly, a comparison of the lifetimes of Li$^+$-F$^-$, Li$^+$-Cl$^-$, and Li$^+$-I$^-$ ion pairs in the eutectic mixture shows that $\tau_{LiF} > \tau_{LiCl} > \tau_{LiI}$, which is exactly the opposite of the lifetimes in the pure compounds, demonstrating that ion pairing is both

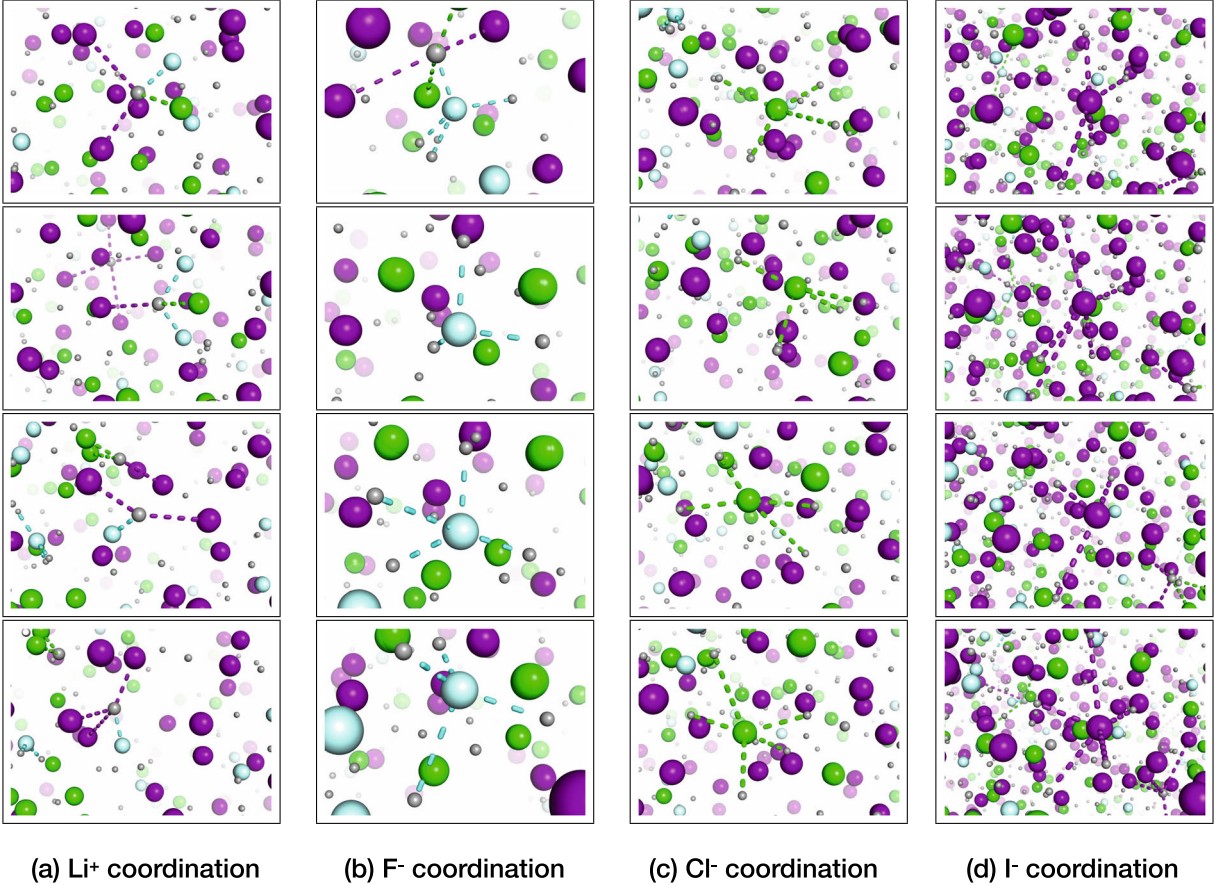

**(a) Li⁺ coordination**    **(b) F⁻ coordination**    **(c) Cl⁻ coordination**    **(d) I⁻ coordination**

**Fig. 3 Snapshots of LiF-LiCl-LiI$_{eut}$ at 1200 K. a** Li⁺ coordination, **b** F⁻ coordination, **c** Cl⁻ coordination, and **d** I⁻ coordination. In gray: Li⁺, light blue: F⁻, green: Cl⁻, and purple: I⁻. For visibility, only a few selected coordination spheres are indicated. Ionic bonds are indicated as dashed lines for the first coordination shell; the bonds were selected as distances that are smaller than the first minima in the respective cation–anion radial distribution functions.

quantitatively and qualitatively different in mixtures. The lifetimes are affected both by temperature and the electric field strength; at higher temperatures and larger electric field strengths, the lifetimes get shorter. However, the observed qualitative trends that are shown in Table 3 remain unchanged and similar trends are observed for the binary and other ternary mixtures (Supplementary Note 6 and Supplementary Table 7). Due to the high molar fraction of I⁻ (~60% of the anions) in LiF-LiCl-LiI$_{eut}$ and the shorter ion-pair lifetime of Li⁺-I⁻, this allows for the formation of isolated short-lived and overall positively charged Li-anion clusters, i.e., Li$_n$F$^{(n-1)}$ and Li$_n$Cl$^{(n-1)}$ (Fig. 3b–d). The formation of such clusters is in line with F⁻ and Cl⁻ drifting in the same direction as Li⁺. The analysis of the lifetime and the Gibbs free energies of activation $\Delta^{\ddagger}G$ of the bond breaking are consistent with the trends seen in CNs and bond distances[17].

## Discussion

In this contribution, the microscopic origin of the conductivity in LiF$_\alpha$Cl$_\beta$I$_\gamma$ systems is explored by directly investigating the mobility of the ions using equilibrium and non-equilibrium computer simulations. A key finding is that a high conductivity can coincide with a low Li⁺ electrical mobility as the system has a high number density. This has implications for the design of electrolyte materials, because optimizing the conductivity is one of the key objectives for the design of a liquid metal battery. We suggest that it is in fact more reasonable to increase the conductivity via an increase in the number density, i.e., by changing the LiF$_\alpha$Cl$_\beta$I$_\gamma$ system's composition; even though this is accompanied by a drop

in the electrical mobility (see Supplementary Note 1 and Supplementary Fig. 2). In contrast, an increase of the conductivity via an increase in the ions mobility, i.e., by increasing the temperature, leads to lowering of the number density, counteracting the conductivity gain due to temperature. As shown in Supplementary Fig. 2, by changing the system's composition at the same temperature, the conductivity can be changed more significantly than by a change in temperature of the same system. Other properties, such as the melting point, may be important for the optimization of an electrolyte as well and a multi-dimensional property screen may be needed.

Molten alkali halides are in general considered to be fully ionized. However, in our non-equilibrium simulations of binary and ternary alkali halide mixtures, we observe that the lighter anion(s) drift(s) along with Li⁺ towards the negatively charged electrode. This can be explained by the notion that the conservation of momentum holds for an overall neutral (closed) system when an electric field is applied, with the partial conductivities following Sundheim's golden rule. In mixtures with one cation and multiple anions, the cation will drift towards the negatively charged electrode, whereas the anion that binds weakest to the cation will move towards the positively charged electrode. The other anion(s) may move in either direction or barely at all (Table 2). In a mixture with one anion and multiple cations[14], the reverse is true.

Both structure and dynamics in the LiF-LiCl-LiI$_{eut}$ mixture differ significantly from the pure compounds. As a result, the thermodynamic equilibria are changed in a way that allows for the formation of transient overall positively charged Li$_n$F$^{(n-1)}$

and Li$_n$Cl$^{(n-1)}$ clusters that dissociate on a ps-timescale, which is in line with the ionic bond character of alkali halides. The lifetime of such clusters is dependent on the composition of the melt, its temperature and the applied electric field strength, but remains in the ps-timescale. The formation of those clusters agrees with the lighter anions drifting in the same direction as the Li$^+$ cation on average in an electric field. Similar trends are observed for the binary eutectic and other ternary mixtures, but not for the pure compounds. This behavior is revealed by the application of non-equilibrium simulations that allow to gain real insight on the microscopic origin of conductivity by directly observing the effect of an electric field on the dynamics of the ions.

Molecular simulations provide a link between microscopic insight into structure and dynamics of these systems, and macroscopic properties such as conductivity that can be analyzed in an objective way. This is because coordinates and velocities are directly available for analysis, the frame of reference is well-defined in a periodic simulation box, and the system under study can readily be manipulated, for instance by applying an electric field. We are therefore confident that the ongoing debate in the literature on ion association in molten salts and ILs, and its effect on conductivity[6,8,10,11,13–15], can be advanced by quantitative analysis of ion-pair lifetimes and partial conductivities, as provided here for lithium halide melts. This microscopic insight then allows for optimizing of relevant properties, such as, e.g., partial conductivity of lithium, and to further the understanding of its contribution to macroscopic properties, such as the total conductivity.

## Methods

MD simulations were carried out using a recently published classical force field for alkali halides employing explicit polarization and distributed charges[16]. The model has been applied and validated in a number of further applications to study liquid structure[18] and melting points[29]. In a further study, a link was made between structure, dynamics, and thermodynamics in molten salts[17]. The GROMACS simulation software (version 4.6.7)[57] was used applying tabulated Coulomb (coulomb-type: PME-User) and van der Waals potentials (vdw-type: User), using tables with a spacing of 0.5 pm and double precision. The integration step size was 1 fs and the cutoff length for the Coulomb and van der Waals interactions was equal to approximately half of the box size. A dispersion correction for energy and pressure was applied after the cutoff. For the polarizable force field, the convergence tolerance for the shell minimization was set to 0.1 kJ mol$^{-1}$ nm$^{-1}$[58]. All simulations were performed using periodic boundary conditions. Boxes with pure salts contained 1000 ions and boxes with binary or ternary mixtures contained 2000 ions. All boxes were equilibrated by running constant pressure simulations (i.e., with an isothermal-isobaric ensemble) using the Berendsen isotropic pressure coupling ($\tau_P$ 10 ps)[59] and for the temperature coupling the velocity rescaling thermostat ($\tau_T$ 0.1 ps)[60]. All melts were equilibrated for 2 ns at the respective temperature (see Supplementary Note 1, Supplementary Tables 1 and 2, and for comparison a ternary plot of the experimental melting points is provided in Supplementary Note 7 and Fig. 10); their energy and density were checked for convergence. In continuation, a 6 ns constant volume (i.e., a canoical ensemble (NVT)) simulation was run and used to perform the structural and dynamic analyses. The diffusion coefficients were calculated using the mean square displacement (msd) method. In continuation to those simulations, 500 ps NVT simulations were run with a time step of 1 fs; the position and velocity were saved at every step and were used for the calculation of the velocity and current ACFs. By means of the velocity autocorrelation (vac) function, the diffusion coefficients were determined in addition to the ones determined via the msd method (Supplementary Methods and Supplementary Eqs. (5) and (6)).

The conductivity was evaluated via four different methods (NE, GK, EH, and by means of non-equilibrium electric field simulations) for which the mathematical equations are provided in the Supplementary Methods section (Supplementary Eqs. (1)–(4)). The NE method uses the diffusion coefficients that were determined using the msd and vac method (see Supplementary Note 2 and Supplementary Figs. 3 and 4). In addition, we evaluated the conductivity using the 500 ps NVT run using the more accurate GK and the EH method. For the GK conductivity, we evaluated the running integral of the current ACF up to 4 ps; the conductivity was calculated as the average value between 1 and 4 ps (see Supplementary Note 2 and Supplementary Fig. 5). For the EH conductivity, the slope of the translation dipole moment was evaluated from 0 to 10 ps where a reasonable agreement was found with the GK evaluation method. For a visual comparison of the different conductivity results, the reader is referred to Supplementary Note 3 and

Supplementary Figs. 6–8. For comparison, a ternary plot of the experimental conductivity values is provided in the Supplementary Note 7 and Supplementary Fig. 10.

For the non-equilibrium simulations (2 ns, NVT), an external electric field (0.1, 0.2, 0.3, and 0.4 V nm$^{-1}$) was applied to the different equilibrated simulation boxes and the drift velocity in the electric field direction was evaluated. The electric field is modeled by applying an extra force in x-direction on each particle with $F_i = q_i E_x$. In comparison to a real system with physical electrodes, also in a simulation, there may exist screening effects. Because of asymmetric distribution of ions in an electric field Fig. 3, a counter-field is generated[61], which may affect the electrical mobility. However, in contrast to a real system, there is no interface between the molten salt and the electrode in the simulation. Therefore, the simulations are representative of the bulk of a system that is exerted to an electric field.

The drift velocity $v_d$ is computed as the average velocity over all ions and the whole simulation length in the electric field direction. The velocity of single ions was inspected over the simulation length and no transient regime in the beginning of the simulation was found, because the system, i.e., the ion velocities, adapted to the field instantaneously. Using a linear regression analysis, the ion mobility was determined from the slope in the plots drift velocity vs. electric field strength ($b_{EF} = v_d/E$). The parameters for the polarizable force field for alkali halides (Wang-Buckingham) are reported in Walz et al.[16] and are available from http://virtualchemistry.org[62].

## Data availability

The datasets generated and analyzed during the current study are available from the authors on reasonable request. In the Supplementary Information, section Supplementary Methods, we provide equations for conductivity evaluation methods (Nernst–Einstein, Green–Kubo, Einstein–Helfand, and Electric field) and equations for the diffusion coefficient evaluation methods (Einstein and Green–Kubo). The following sections contain: Supplementary Note 1: tables summarizing the evaluated diffusion coefficients and conductivities, and a figure showing the Green–Kubo conductivity vs. the applied temperature; Supplementary Note 2: velocity ACFs and current ACFs; Supplementary Note 3: plots showing diffusion coefficients and conductivity; Supplementary Note 4: a table listing mobilities and partial conductivities, the parametric expression to approximate the conductivity data, a table comparing simulated vs. calculated conductivity, and transference numbers; Supplementary Note 5: radial distribution functions (cation–anion); Supplementary Note 6: details on the evaluation of bond distances, CNs, lifetimes, and Gibbs free energy of activation of bond breaking; and Supplementary Note 7: ternary plots of the experimental melting points and conductivities. Furthermore, two Supplementary Movies are provided. Both movies show the movement of selected ions in molten LiF-LiCl-LiI$_{eut}$ at 1200 K with an applied electric field (0.4 V nm$^{-1}$) for 1 ns (Movie 1) and 50 ps (Movie 2, here also the coordination sphere is indicated).

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

## Acknowledgements

The Swedish research council is acknowledged for a grant of computer time (SNIC2018-2-42, SNIC2019-2-32, and SNIC2019-35-63) through the High Performance Computing Center North in Umeå, Sweden. Funding from eSSENCE - The e-Science Collaboration (Uppsala-Lund-Umeå, Sweden) is gratefully acknowledged.

## Author contributions

M.M.W. and D.v.d.S. designed the work. M.M.W. conducted and analyzed the MD simulations. D.v.d.S. contributed to the analysis of the MD simulations. M.M.W. and D.v.d.S. interpreted the data. M.M.W. drafted the manuscript and D.v.d.S. substantively revised the manuscript.

## Funding

## Competing interests

The authors declare no competing interests.
