## [Peer Review File · Communications Chemistry]

Reviewers' comments:

Reviewer #1 (Remarks to the Author):

Walz and van der Spoel investigate ion transport in molten salts (mixtures of LiF, LiCl, and/or LiI) through molecular dynamics simulations. Interactions are modeled using a classical potential which includes a description of electronic polarization, previously reported by the authors. Transport properties are computed using a combination of both equilibrium and non-equilibrium methods. The main findings are the following:

- *) while the total ionic conductivity is dominated by the Li⁺ contribution (table S3), high conductivity does not necessarily rely on high Li⁺ mobility (figure 1b);
- *) low conductivity in the phase diagram surroundings of the LiF-LiCl-LiI eutectic mixture is due to a combination of low Li⁺ density, and high ion-ion correlations (figure 2);
- *) in binary and ternary mixtures of lithium halides, on average, the lighter anion(s) drift along Li⁺ (table S3), which is correlated to variations in the relaxation time of ion pairs and in coordination numbers (table S6).

The manuscript is well structured. The findings are clear and supported by convincing evidence, especially regarding the range of methods used to determine transport properties. Results are thoroughly discussed and compared with recent reports, including both experimental (e-NMR) and computational (molecular dynamics) work (e.g. refs 14 and 40). To me, the findings of positive mobility for anions and > 1 Li⁺ transference number in molten salts constitute an interesting and timely development. I'd recommend publication in Communications Chemistry, once a few points - mostly minor - are addressed.

1) More details should be included regarding the non-equilibrium method:

a) How do the authors model the electric field? I suspect it is by adding a force to all particles, proportional to their partial charges. If it is the case, this should be clearly stated, as well as the implications of such a model (no screening effects compared to a model with physical electrodes, for instance).

b) How exactly is the drift velocity computed? Is there a transient regime during the 2 ns long non-equilibrium simulations, when the field is switched on?

2) Variations in the lifetime of ions pairs and in coordination numbers, as well as the presence of positively charged aggregates composed of Li⁺ and anions (some examples shown in figure 3), seem to explain the observed positive mobility of some anions. While this is convincing, it remains an indirect evidence. The authors should either state that, or perform more thorough analyses. At the very least, a cluster population analysis such as the one reported in ref. 42 (figure 4) could be included. Going further, from the non-equilibrium trajectories, environment-dependent (i.e. cluster-dependent) ionic mobilities could be computed simply by partitioning the drift velocities according to the local instantaneous coordination.

3) In figure 1c, the x-axis range could be chosen differently. This would allow to label all temperatures. I understand the range is chosen such that it is the same as in panel d; an axis break could be included from $x = 25$ to 45 ions/nm³, with a linear regression in panel d to emphasize the point made.

4) page 2, line 32: "which can lead, ...". To me, it seems like the way this paragraph is written overall suggests that temperature and conductivity are not strongly positively correlated. Again, to me, the conductivity maximum in semicovalent molten salts mentioned here is the exception and is anomalous, and most of the time, an Arrhenius or VTF behavior is observed. Furthermore, I don't think a maximum in conductivity is observed in lithium halide mixtures (see e.g. 10.1016/j.jpowsour.2006.01.014).

5) page 2, line 31: "However, as a result the liquid expands, the counter ions get closer together (ref 18)". The point made in ref 18 is not extremely clear to me. For instance, the coordination number variations reported are smaller than their uncertainty. It is much clearer in ref. 17, from the same authors.

6) page 12, line 204: "both" cannot be used for more than two items.

7) I would suggest redesigning figure 3. Instead of having a panel for each ion, with all ions represented in each panel, I'd suggest having a few panels per ion type, each panel centered on one coordination sphere only, representing only the "bonded" ions selected using the distance criterion mentioned. This way, more diverse coordination environments can be presented, and in a clearer way.

Arthur France-Lanord

Reviewer #2 (Remarks to the Author):

The major claims of the paper are that there is a decoupling of conductivity and electrical mobility, that to increase conductivity it makes more sense to increase ion concentration (number density). This causes one to take a step back to think about how to optimize transport properties in a molten salt. This reviewer would like to see a parametric expression of how the expected increase in conductivity with increasing temperature according to Arrhenius activation is offset by the decrease in number density. A compact ratio that proves that per unit increase in temperature there is a decrease in conductivity would be helpful in quantitatively making the point here. On p. 9 the paper states that under an electric field Li^+ migrates to the negative electrode as do the lighter anions, F^{sup} and Cl^{sup} . How is this possible? Where are electrostatics? On p. 2, line 19, there is reference to ion pairing and clustering. Does this mean that there can be formation of covalently bonded anionic complexes around the Li^+ , e.g., $\text{LiF}_2^{\text{sup}}$? They would head towards the positive electrode, would they not? On p. 11, lines 183-184, again there is reference to negatively charged clusters, but these head towards the positive electrode. The writing lacks clarity and falls short of illuminating what the deep insight is that sheds new light on an old problem.

Reviewer #3 (Remarks to the Author):

The authors explore a method for determination of ionic electrical mobilities based on non-equilibrium computer simulations. Partial conductivities are determined as a function of system composition and temperature from simulations of molten LiFAlCl_4 .

However, the most bizarre point of this paper is, as summarized in Table 2, that the ion mobilities and ion conductivities by EF and D are almost inverse signs. The authors stated that μ_{EF} determined by means of applying an external electric field includes ion-ion interactions, on the other hands μ_{D} determined from diffusion coefficients ignores these effects.

I disagree with this explanation. The mobility estimated from the diffusion coefficient D also includes the interaction between ions. The results for EF and D should match. Such discrepancies could occur if the external electric field is improperly too large. I don't think the authors have discovered a new phenomenon that denies the knowledge of textbooks.

Reviewers' comments: in black

Answers: in blue

General remark:

We thank all reviewers for their valuable comments and suggestions. All comments are answered and changes to the text are highlighted in bold in the revised manuscript, and for clarity also added to this document.

Reviewer #1 (Remarks to the Author):

Walz and van der Spoel investigate ion transport in molten salts (mixtures of LiF, LiCl, and/or LiI) through molecular dynamics simulations. Interactions are modeled using a classical potential which includes a description of electronic polarization, previously reported by the authors. Transport properties are computed using a combination of both equilibrium and non-equilibrium methods. The main findings are the following:

- *) while the total ionic conductivity is dominated by the Li⁺ contribution (table S3), high conductivity does not necessarily rely on high Li⁺ mobility (figure 1b);
- *) low conductivity in the phase diagram surroundings of the LiF-LiCl-LiI eutectic mixture is due to a combination of low Li⁺ density, and high ion-ion correlations (figure 2);
- *) in binary and ternary mixtures of lithium halides, on average, the lighter anion(s) drift along Li⁺ (table S3), which is correlated to variations in the relaxation time of ion pairs and in coordination numbers (table S6).

The manuscript is well structured. The findings are clear and supported by convincing evidence, especially regarding the range of methods used to determine transport properties. Results are thoroughly discussed and compared with recent reports, including both experimental (e-NMR) and computational (molecular dynamics) work (e.g. refs 14 and 40). To me, the findings of positive mobility for anions and > 1 Li⁺ transference number in molten salts constitute an interesting and timely development. I'd recommend publication in Communications Chemistry, once a few points - mostly minor - are addressed.

Answer:

We thank the reviewer for carefully reading our manuscript, valuable suggestions and constructive criticism. We address the reviewer's comments below.

- 1) More details should be included regarding the non-equilibrium method:
 - a) How do the authors model the electric field? I suspect it is by adding a force to all particles, proportional to their partial charges. If it is the case, this should be clearly stated, as well as the implications of such a model (no screening effects compared to a model with physical electrodes, for instance).

Answer:

In our simulations a static electric field was applied in the x-direction using the .mdp option available in GROMACS (see <http://manual.gromacs.org/current/reference-manual/special/electric-fields.html>) using electric field strengths 0.1, 0.2, 0.3 and 0.4 V/nm. The reviewer is correct in the assumption that each particle is exerted to an added force ($F = q * E$), and there are differences if one compares our simulation to a real system with physical electrodes.

Regarding the mentioned screening effects, in fact, even in a simulation there may exist screening effects due to the asymmetric distribution of ions surrounding the ion that is exerted to an electric field. However, we do not have an interface between the salt melt and the electrodes, and due to periodic boundary conditions there is a free flow of ions. Therefore, our simulations are representative of the bulk of a system that is exposed to an electric field.

We added the following text in the supporting information to provide more details on the applied electric field:

“The electric field is modeled by applying an extra force in x-direction on each particle with $F_i = q_i E_x$. In comparison to a real system with physical electrodes, also in a simulation there may exist screening effects. Because of asymmetric distribution of ions in an electric field (Figure 3) a counter-field is generated⁹ that may affect the electrical mobility. However, in contrast to a real system, there is no interface between the molten salt and the electrode in the simulation. Therefore, the simulations are representative of the bulk of a system that is exerted to an electric field.”

b) How exactly is the drift velocity computed? Is there a transient regime during the 2 ns long non-equilibrium simulations, when the field is switched on?

Answer:

The drift velocity is computed using the GROMACS module “gmx traj” (see <http://manual.gromacs.org/documentation/2019-current/onlinehelp/gmx-traj.html>). Using the trajectory files (from the simulations where an external electric field was applied), the average velocity of each ion is calculated in x, y and z direction over the full length of the simulation. The velocity of single ions was monitored and we do not find a visible “transient regime” when the field is switched on.

In a next step the module “gmx analyze” is used to calculate the average velocity of all the ions in x, y and z direction:

(see <http://manual.gromacs.org/documentation/2019-current/onlinehelp/gmx-analyze.html>). The drift velocity is the average velocity (over all involved ions and frames) in the electric field direction, in our case the x-direction. We also tested to calculate the average velocities in the x, y and z direction skipping the first 500 ps of the simulation, and the average velocities stay the same within the margin of error.

We added the following text in the supporting information to include more details on how the drift velocities are calculated:

“The drift velocity is computed as the average velocity over all ions and the whole simulation length in the electric field direction. The velocity of single ions was inspected over the simulation length and no transient regime in the beginning of the simulation was found, because the system, i.e. the ion velocities, adapted to the field instantaneously.”

2) Variations in the lifetime of ions pairs and in coordination numbers, as well as the presence of positively charged aggregates composed of Li⁺ and anions (some examples shown in figure 3), seem to explain the observed positive mobility of some anions. While this is convincing, it remains an indirect evidence. The authors should either state that, or perform more thorough analyses. At the very least, a cluster population analysis such as the one reported in ref. 42 (figure 4) could be included. Going further, from the non-equilibrium trajectories, environment-dependent (i.e. cluster-dependent) ionic mobilities could be computed simply by partitioning the drift velocities according to the local instantaneous coordination.

Answer:

We agree with the reviewer that this is an indirect evidence, and we modified the text in the manuscript accordingly.

“Ion pairs and clusters are formed, and the Li⁺ **seem to** effectively drag along the lighter anions.”

“[...] this allows for the formation of isolated short-lived and overall positively charged Li-anion clusters, i.e. Li_nF⁽ⁿ⁻¹⁾ and Li_nCl⁽ⁿ⁻¹⁾ (Fig. 3). **The formation of such clusters is in line with F⁻ and Cl⁻ drifting** in the same direction as Li⁺.”

“**The formation of those clusters agrees with** the lighter anions **drifting** in the same direction as the Li⁺ cation on average in an electric field.”

3) In figure 1c, the x-axis range could be chosen differently. This would allow to label all temperatures. I understand the range is chosen such that it is the same as in panel d; an axis break could be included from x = 25 to 45 ions/nm³, with a linear regression in panel d to emphasize the point made.

Answer:

We understand the reviewer’s point that it would be nice to label all of the data points. However, we think that the figure would get too crowded. We would like to point out that the data presented in Figure 1c is provided in Table S2, and we modified the figure caption in the following way to add additional information about the applied temperatures. Figure 1 d was modified according to the reviewer’s suggestion.

“Figure 1: Conductivity vs. Li⁺ mobility (a and b) and vs. Li⁺ number density (c and d). (a and c) At different temperatures for LiF-LiCl-LiI_{eut} (**623, 723, 773, 1000 and 1200 K**) and for LiF- LiCl-LiI_{400/450} (**773 and 1200 K**); a higher mobility and lower number

density correspond to a higher temperature. [...] Numerical details are provided in Table S1 and S2."

4) page 2, line 32: "which can lead, ...". To me, it seems like the way this paragraph is written overall suggests that temperature and conductivity are not strongly positively correlated. Again, to me, the conductivity maximum in semicovalent molten salts mentioned here is the exception and is anomalous, and most of the time, an Arrhenius or VTF behavior is observed. Furthermore, I don't think a maximum in conductivity is observed in lithium halide mixtures (see e.g. 10.1016/j.jpowsour.2006.01.014).

Answer:

Yes, we agree with the reviewer that a maximum in the conductivity is the exception, and we merely wanted to state that such behavior exists. We modified this paragraph to make this clearer (changes in bold):

"An increase of the temperature is an obvious manner to increase the velocity of the ions **which in turn increases the conductivity in general**. However, **with increasing temperature the overall liquid expands, while the counter ions get closer together**,¹⁸ and the energy barrier for hopping to the next neighbour increases,^{9,12,17} which can lead in **certain cases even** to a maximum in the conductivity as a function of temperature **such as for example in the case of semicovalent molten halides (e.g. BiCl₃)**.^{9,12,13} "

5) page 2, line 31: "However, as a result the liquid expands, the counter ions get closer together (ref 18)". The point made in ref 18 is not extremely clear to me. For instance, the coordination number variations reported are smaller than their uncertainty. It is much clearer in ref. 17, from the same authors.

Answer:

It is true that the trends of interionic distance, coordination number and ion-pair lifetime with temperature are more pronounced in ref. 17 than in ref. 18 as the investigated temperature range is much larger in ref. 17. However, we think that Fig. 4 in ref. 18 illustrates very nicely the effect of the counter ions getting closer together with increasing temperature, while the overall liquid is expanding based on that the interionic distances getting larger from the second coordination shell onwards. To follow the reviewer's suggestion, we added ref. 17 as well.

6) page 12, line 204: "both" cannot be used for more than two items.

Answer:

The reviewer is correct, and we thank the reviewer for carefully reading the manuscript. The word "both" is the remainder of an older version of the manuscript. It has now been removed.

7) I would suggest redesigning figure 3. Instead of having a panel for each ion, with all ions represented in each panel, I'd suggest having a few panels per ion type, each panel centered on one coordination sphere only, representing only the "bonded" ions selected using the distance criterion mentioned. This way, more diverse coordination environments can be presented, and in a clearer way.

Answer:

We thank the reviewer for this good idea and we modified Fig. 3 according to the reviewer's suggestion.

Reviewer #2 (Remarks to the Author):

The major claims of the paper are that there is a decoupling of conductivity and electrical mobility, that to increase conductivity it makes more sense to increase ion concentration (number density).

Answer:

By stating “that to increase conductivity it makes more sense to increase ion concentration (number density)”, we simply wanted to point out that the conductivity can be increased more significantly via the number density than it can be by the temperature. The following figure might illustrate this more clearly. We added this figure also to the supporting information. In this figure the conductivity is plotted vs. the temperature. As it can be seen that the conductivity indeed increases with increasing temperature (due to the increase in the mobility; the number density decreases). However, it can also be seen that at the same temperature (at 1200 K), by changing the system’s composition, the conductivity can be changed more significantly (due to an increase in number density; the mobility decreases when the number density increases).

Modification to the text:

“We suggest that it is in fact more reasonable to increase the conductivity via an increase in the number density, i.e. by changing the $\text{LiF}_\alpha\text{Cl}_\beta\text{I}_\gamma$ system’s composition; even though this is accompanied by a drop in the electrical mobility (see Figure S2). [...] As shown in Figure S2, by changing the system’s composition at the same temperature, the conductivity can be changed more significantly than by a change in temperature of the same system.”

This causes one to take a step back to think about how to optimize transport properties in a molten salt. This reviewer would like to see a parametric expression of how the expected increase in conductivity with increasing temperature according to Arrhenius activation is offset by the decrease in number density. A compact ratio that proves that per unit increase in temperature there is a decrease in conductivity would be helpful in quantitatively making the point here.

Answer:

We did not write that there is a decrease in conductivity with an increase in temperature for the investigated systems, as is the case for semicovalent molten halides at certain temperatures. The conductivity increases with increasing temperature. We added also the following text to make this clearer:

“An increase of the temperature is an obvious manner to increase the velocity of the ions **which in turn increases the conductivity.**”

The point that we tried to make is simply that with an increase in temperature the mobility contributes positively to the conductivity, while the number density that decreases contributes negatively according to the equation $\sigma_i = \rho_{N,i} q_i b_i$. But “the expected increase in conductivity with increasing temperature ... is” **not** “offset by the decrease in number density”; it is only “counteracted” as we wrote in our manuscript. In the same way is the increase of the conductivity by an increase in the number density “counteracted” by the decrease in the mobility, as we also noted.

We appreciate the reviewer’s idea to represent our conductivity data by means of a parametric expression. It is an interesting idea to derive a such an expression that only depends on the composition and the temperature, and we added the following part in the main manuscript:

“In analogy to the empirical equation that was derived by Redkin et al.³³ to describe the conductivity in dependence of temperature and composition, we can interpolate our conductivity data with the following parametric expression:

$$\ln(\sigma_{x_i}, T) = \alpha - (\beta/T) + (\gamma / (x_{LiF}\delta + x_{LiCl}\epsilon + x_{LiI}\zeta)) + (\eta / (x_{LiF}\delta + x_{LiCl}\epsilon + x_{LiI}\zeta)^2) \quad (1)$$

with T being the melt’s temperature, x_i being the molar fraction of the melt’s components and α , β , γ , δ , ϵ , ζ and η being the parameters that have been fitted to reproduce the conductivity data. Using this parametric expression, all simulated conductivity data (for different compositions and temperatures) can be reproduced with an RMSD value of 0.2 S/cm. For the calculated conductivity values and the fitted parameters the reader is referred to the supporting information (Table S4).”

And the following part in the supporting information:

“With the following parametric expression the conductivity data can be approximated in dependence of the salt’s composition (x_i being the molar fraction) and its temperature T:

$$\ln(\sigma_{x_i}, T) = \alpha - (\beta/T) + (\gamma / (x_{LiF}\delta + x_{LiCl}\epsilon + x_{LiI}\zeta)) + (\eta / (x_{LiF}\delta + x_{LiCl}\epsilon + x_{LiI}\zeta)^2) \quad (6)$$

with $\alpha = 2.145$, $\beta = 1048.61$, $\gamma = 655.73$, $\delta = 592.51$, $\epsilon = 1290.25$, $\zeta = 8294.43$ and $\eta = 1379.22$.

Table S4: Comparison of the conductivity values determined from the electric field simulations vs. conductivity values (in S/cm) calculated using the parametric expression.

salt	T / K	$x_{LiF} - x_{LiCl} - x_{LiI}$	σ_{EF}	σ_{calc}
LiF	1200	1 - 0 - 0	10.86	10.82
LiCl	1200	0 - 1 - 0	6.35	5.93
LiI	1200	0 - 0 - 1	3.84	3.86
LiF-LiCl _{eut}	1200	0.305 - 0.695 - 0	6.19	6.56
LiF-LiI _{eut}	1200	0.165 - 0 - 0.835	3.53	3.91
LiCl-LiI _{eut}	1200	0 - 0.346 - 0.654	4.17	3.99
LiF-LiCl-LiI _{eut}	1200	0.117 - 0.291 - 0.592	4.05	4.03
LiF-LiCl-LiI ₄₀₀	1200	0.2 - 0.4 - 0.4	4.13	4.21
LiF-LiCl-LiI ₄₅₀	1200	0.25 - 0.55 - 0.2	4.69	4.63
LiF-LiCl-LiI _{eut}	773	0.117 - 0.291 - 0.592	2.63	2.49
LiF-LiCl-LiI ₄₀₀	773	0.2 - 0.4 - 0.4	2.54	2.60
LiF-LiCl-LiI ₄₅₀	773	0.25 - 0.55 - 0.2	2.81	2.86
LiF-LiCl-LiI _{eut}	623	0.117 - 0.291 - 0.592	1.66	1.79
LiF-LiCl-LiI _{eut}	723	0.117 - 0.291 - 0.592	2.17	2.26
LiF-LiCl-LiI _{eut}	1000	0.117 - 0.291 - 0.592	3.61	3.38

????

????”

On p. 9 the paper states that under an electric field Li⁺ migrates to the negative electrode as do the lighter anions, F⁻ and Cl⁻. How is this possible? Where are electrostatics?

Answer:

That Li⁺ and the lighter anions F⁻, and in certain cases also Cl⁻, migrate towards the negative electrode is derived from the drift velocities (that are direct observables) and from this we derived the mobility (with $b_{EF} = v_d/E$, see also SI, Fig. S1). The determined partial conductivities (that are calculated from the mobilities) are indeed in line with Sundheim’s universal golden rule that is derived from the generalized Drude theory as a law of motion under an electric field (see e.g. ref. 34: Sundheim,

1956, J.Phys.Chem. 60 and ref. 35: Tamaki et al., 2020, IntechOpen, “*Electrical Conductivity of Molten Salts and Ionic Conduction in Electrolyte Solutions*”). This supports that our observation indeed agrees with “electrostatics”.

Furthermore, the observation of the formation of short-lived overall positively charged clusters with the composition of $\text{Li}_n\text{F}^{(n-1)}$ and $\text{Li}_n\text{Cl}^{(n-1)}$ is in line with the average drift direction of F^- , and in certain cases of Cl^- , towards the negatively charged electrode.

To make our text clearer in this point, we added the text:

“This observation that seems at a first glance to defy the simplistic picture of independent ions in an electric field is in fact in line with electrostatics as the ions are strongly interacting with each other. The average drift direction of the lighter anions is in agreement with Drude theory (see below) and in line with the formation of overall positively charged short-lived clusters (see below).”

On p. 2, line 19, there is reference to ion pairing and clustering. Does this mean that there can be formation of covalently bonded anionic complexes around the Li^+ , e.g., LiF_2^- ? They would head towards the positive electrode, would they not?

Answer:

In general the formation of “**covalently bonded**” complexes (such as the suggested $(\text{LiF}_2)^-$) is not anticipated in alkali halide systems, and usually alkali halides are described as fully **ionized** systems. As written on p.14, line 257, the observation is the formation of very **short-lived** transient (due to the strong **ionic bond character**) complexes such as e.g. $(\text{Li}_n \text{F})^{(n-1)}$ that are overall positively charged, and therefore migrating towards the negatively charged electrode.

We modified the text in the following way to make this clearer:

“[...] formation of transient overall positively charged $\text{Li}_n\text{F}^{(n-1)}$ and $\text{Li}_n\text{Cl}^{(n-1)}$ clusters that dissociate on a ps-timescale, which is in line with the ionic bond character of alkali halides. The lifetime of such clusters is dependent on the composition of the melt, its temperature and the applied electric field, but remains in the ps-timescale.”

On p. 11, lines 183-184, again there is reference to negatively charged clusters, but these head towards the positive electrode.

Answer:

In those lines we cite the work from Gouverneur et al. that have studied ionic liquids with the composition $\text{LiTFSA}/\text{EmimTFSA}$ and $\text{LiBF}_4/\text{EmimBF}_4$.

In our study we investigate mixed systems with the **same cation** (Li^+) and various anions (F^- , Cl^- , and I^-), whereas Gouverneur et al. studied mixed systems with the **same anion** (TFSA^- and BF_4^-) and various cations (Li^+ and Emim^+). In Gouverneur’s case the observation is the formation of **negatively charged clusters** ($[\text{Li}(\text{TFSA})_2]^-$ or $[\text{Li}(\text{BF}_4)_3]^{2-}$) that migrate towards the positive electrode, while in our cases the

formation of short-lived **positively charged clusters** is observed that migrate towards the negative electrode. The opposite effect (comparing our with Gouverneur's work) is due to the fact that in Gouverneur et al. study systems with the same anion, while in our case mixed systems with the same cation were studied.

We modified and added the following text to make this clearer:

"Gouverneur et al. studied Li-salts dissolved in different ionic liquids (ILs) (LiTfSA/EmimTfSA and LiBF₄/EmimBF₄) using [...]"

"The observation that in our study anions drift in the "wrong" direction, whereas in Gouverneur's study the cation does, is based on the fact that here mixed systems with the same cation are studied, while Gouverneur et al. investigate mixed systems with the same anion."

The writing lacks clarity and falls short of illuminating what the deep insight is that sheds new light on an old problem.

Answer:

We added the following text in the conclusion in order to clarify and to illuminate our contribution to the understanding of conductivity.

"The formation of those clusters is in line with the lighter anions drifting in the same direction as the Li⁺ cation on average in an electric field. [...] This behavior is revealed by the application of non-equilibrium simulations that allow to gain real insight on the microscopic origin of conductivity by directly observing the effect of an electric field on the dynamics of the ions."

Reviewer #3 (Remarks to the Author):

The authors explore a method for determination of ionic electrical mobilities based on non-equilibrium computer simulations. Partial conductivities are determined as a function of system composition and temperature from simulations of molten LiF_αCl_βly.

However, the most bizarre point of this paper is, as summarized in Table 2, that the ion mobilities and ion conductivities by EF and D are almost inverse signs. The authors stated that bEF determined by means of applying an external electric field includes ion-ion interactions, on the other hands bD determined from diffusion coefficients ignores these effects.

I disagree with this explanation. The mobility estimated from the diffusion coefficient D also includes the interaction between ions. The results for EF and D should match. Such discrepancies could occur if the external electric field is improperly too large. I don't think the authors have discovered a new phenomenon that denies the knowledge of textbooks.

Answer:

It is well known that **conductivity values that are estimated from diffusion coefficients neglect ion cross-correlation effects**, which is why it has been pointed out by numerous authors that the *Nernst-Einstein* (NE) equation is only a means to **estimate** the conductivity (and frequently so-called Haven ratios are determined that quantify the deviation of the NE conductivity from the real conductivity). More accurate approaches, such as the *Green-Kubo* or *Einstein-Helfand* method, should be used that account for ion cross-correlation effects (see e.g. ref. 42: Harris, 2010, *J.Phys.Chem.B*, 114 and ref. 30: Dommert, 2008, *J.Chem.Phys.* 129).

The same is true for the mobility, b_i , (as the mobility and the conductivity are directly correlated, $\sigma_i = \rho_{N,i} q_i b_i$). **The use of diffusion coefficients will give an inaccurate estimate of the electric mobility values due to the fact that they neglect ion cross-correlation effects** (see e.g. ref. 10: Schönhoff, 2018, *PCCP*, 20 and ref. 14: Gouverneur, 2018, *PCCP*, 20). Therefore, it is not surprising that the electrical mobility and conductivity values (determined from the electric field simulations) do not match with the ones estimated using the diffusion coefficients. The observed discrepancies are due to the neglect of ion cross-correlation effects.

In fact, work from Zhang et al. (ref. 23: *ACSMacroLetters*, 2020, 9) investigates this by determining diffusion coefficients D_{ij} , “quantifying the slope of the **long time correlated displacements of the distinct species i and j**”, that are then used to calculate σ_{--} , σ_{++} , and σ_{+-} , that “represent the conductivity contributions arising, respectively, from distinct anion–anion, distinct cation–cation, and cation–anion correlated motions”. Zhang et al. defines here the conductivity as “ $\sigma = \sigma_{-} + \sigma_{+} + \sigma_{-} + \sigma_{++} + 2\sigma_{+-}$ ” with “ $\sigma_{-} = (pe^2/k_B T)x_{-z}^{-2}D_{-}$ and $\sigma_{+} = (pe^2/k_B T)x_{+z}^{-2}D_{+}$ are, respectively, the **conductivity contributions arising from the anions and cations** in the framework in which the ions are **moving in an uncorrelated manner**”. That is $\sigma = \sigma_{-} + \sigma_{+}$ is the Nernst-Einstein (NE) equation, whereas $\sigma_{-} + \sigma_{++} + 2\sigma_{+-}$ is the conductivity contribution that is due to ion cross-correlation effects that are neglected by the NE approach.

The reviewer argues that our electric field strengths might be too large. However, as demonstrated in the supporting information, Figure S1, where the drift velocity is plotted vs. the electric field strength, we see that we are well within the linear response regime. This is also supported by the fact that the conductivity values determined from the non-equilibrium simulations using an external electric field agree with the ones determined from the accurate equilibrium approaches like Green-Kubo and Einstein-Helfand that are well established in literature. Furthermore, the force between two ions that are separated by a distance of 0.2 nm is approx. 3500 kJ/(mol*nm), whereas the force on one ion that is exerted to the strongest applied electric field (0.4 V/nm) corresponds to approx. 40 kJ/(mol*nm). This comparison shows that the internal electrical forces are larger by a factor of approx. 90. Regarding the negative sign of the certain mobility values, we would like to point out that we are not the first authors that observe negative transference numbers (see e.g. ref. 14: Gouverneur, 2018, *PCCP*, 20 and ref. 40: Molinari, 2019, *J.Phys.Chem.Lett.*, 10). Additionally, we would like to point out that the partial

conductivity values, and thus the partial mobility values, fulfill Sundheim's universal golden rule that is derived from the generalized Drude theory as a law of motion under an electric field. The agreement with Green-Kubo theory and Sundheim's golden rule underline the accuracy of our non-equilibrium mobility and partial conductivity calculations.

We would also like to clarify that we did not write that we "discovered a new phenomenon that denies the knowledge of textbooks".

In order to make the parts that have been discussed by the reviewer clearer, we added and modified the text in the following way:

"As explained above, the b_D values neglect ion cross-correlation effects, and b_D equals b_{EF} only in case there are no interactions between the ions, e.g. in very dilute solutions. ¹⁴ **Ion cross-correlation effects can be quantified by evaluating diffusion coefficients D_{ij} from the slope of the long time correlated displacements of the species i and j . Zhang et al. uses D_{ij} to calculate σ_{--} , σ_{++} , and σ_{+-} that represent the conductivity contributions arising from the anion-anion, cation-cation and cation-anion correlated motions, with the total conductivity defined as $\sigma = \sigma_- + \sigma_+ + \sigma_{--} + \sigma_{++} + 2\sigma_{+-}$ where $\sigma_- (= (e^2/k_B T) \rho_- z_-^2 D_-)$ and $\sigma_+ (= (e^2/k_B T) \rho_+ z_+^2 D_+)$ are the conductivity contributions arising from the anions and cations that move in an uncorrelated manner.**²⁴"

REVIEWERS' COMMENTS:

Reviewer #1 (Remarks to the Author):

I'm happy with the revisions made by the authors, and by the answers they gave to my comments.

For the sake of completeness, a few replies:

* On screening effects with an external field model of an electric field: I agree with you. Actually I was thinking about the screening of the electrode's charges (the structuration of the interface), which doesn't have to be captured anyway in your methodology.

* On figure 1d: the suggested linear regression was really to emphasize the linear behavior if you had included the axis break I suggested. Anyway, the figure is clear enough in its current form.

Arthur France-Lanord

Reviewer #2 (Remarks to the Author):

Editorial Note: This reviewer provided no further comments for the authors.

Reviewer #3 (Remarks to the Author):

To conclude first, this time I agree to publish this article. The reason is as follows.

The authors seem to think that the difference in the transport coefficient results between EMD and NEMD is due to the difference in calculation accuracy, mainly because EMD omits the interaction. If they wanted to argue that it was correct, they should have shown that the EMD results, including the amendments, were consistent with NEMD.

But that seems impossible. As mentioned in some of the references, the deviation from the Nernt-Einstein relationship is simply expressed as follows.

$$\sigma = (pe^2 / 2kBT) (D + + D-) (1-\Delta)$$

Here, the correction term Δ is about 0.1-0.01 and does not exceed 1. That is, it is impossible for the correction term to reverse the sign of the main term, which is no longer a correction!

This shows that there is an essential difference between the EMD and NEMD methodologies that is not a matter of computational accuracy. Therefore, I agree to publish this paper and submit it to public discussion on this point.